# Influence of the Fibre Content, Exposure Time, and Compaction Pressure on the Mechanical Properties of Ultraviolet-Cured Composites

**Natalia G. Pérez-de-Eulate [1], Ane Aranburu Iztueta [1], Koldo Gondra [2] and Francisco Javier Vallejo [1,\*]**

[1] IDEKO, Arriaga Industrialdea, 2, E-20870 Elgoibar, Spain; ngutierrez@ideko.es (N.G.P.-d.-E.); anearanburu@hotmail.com (A.A.I.)

[2] GAIKER, Parque Tecnológico, Edif. 202, 48170 Zamudio, Spain; gondra@gaiker.es

[\*] Correspondence: fjvallejo@ideko.es; Tel.: +34-(0)943-748000; Fax: +34-(0)943-743804

**Abstract:** A new process for the impregnation, consolidation, and curing of glass-fibre-reinforced polyester composites was developed to reduce manufacturing costs and secure end properties that compete with other traditional materials. This new process, based on the ultraviolet (UV) curing of prepregs, could be a viable alternative to infusion and other processes. In this paper, we showed that glass fibre composites 3 mm thick could be easily formed using suitable photoinitiating systems. We achieved improved mechanical properties through the application of favourable parameters to traditional manufacturing processes such as hand lay-up and infusion. The prepreg polymerization was monitored by dielectric analysis (DEA), and we evaluated the relationship between the UV radiation exposure time and curing degree. Both the exposure time and compaction pressure affected the fibre content of composites and interlaminar shear strength. Experimental results showed that compaction pressures higher than 4 bar are necessary to increase the mechanical properties of the UV-cured composites. Finally, the properties of the composites manufactured by this new process were compared to the properties of composites manufactured using traditional processes such as hand lay-up and infusion.

**Keywords:** glass-fibre-reinforced composites; impregnation; UV curing prepregs; compaction pressure; degree of curing; infusion process

## 1. Introduction

Composite materials are designed and manufactured for application in several areas such as the aerospace, automotive, civil engineering, and shipbuilding industries. Their key advantages include high mechanical properties, excellent weight/properties balance, and good corrosion behaviour. There is an ongoing research of these materials in different sectors such as the food packing industry, where bionanocomposites have been observed to enhance the mechanical and barrier properties (water and UV) and thermal stability [1,2] of the produced films. Nevertheless, to increase the applications of the composite materials, a better balance between the properties and manufacturing cost is required. For instance, in the automotive sector, middle-sized composite parts applied in internal components, or even in body works, are manually manufactured. These composite parts have a high cost, and many property requirements such as high strength, low weight, long-term durability, high service temperature, or dimensional stability [3–5]. Hand lay-up and spray lay-up processes are used to manufacture these parts due to their versatility and low cost. However, these manual processes present several drawbacks related to the volatile organic components (VOCs) involved, which are

emitted when a high resin content is used. This leads to dangerous working conditions. Usually, handmade composites have relatively low mechanical properties because of the use of inaccurate volumes of resin.

Other typical processing routes for composite parts that could overcome these disadvantages include the infusion process, resin transfer moulding, and compression moulding. It is well known that conventional processes, such as the infusion process, yield better mechanical properties, but at the same time, generate numerous waste materials, such as plastic films, tubes, and sealing tapes, resulting in high costs [4]. On the other hand, other liquid moulding processes, such as resin transfer moulding (RTM), give rise to controlled properties and geometries of the composite parts but require high investments for the equipment and moulds [4]. Compression moulding is also a suitable process for these parts. However, large series are required, because the investment in both the equipment and tooling is high [4].

The easiest way to control the amount of resin in composite processing is to work with previously impregnated materials (prepregs). It is well known that prepregs are widely used in sectors where mechanical requirements are critical, such as in the aerospace sector [3–5]. However, the processing costs are extremely high. This is because working with prepregs has two essential disadvantages, namely, the required storage at low, controlled temperatures ($-20$ °C) to prevent resin curing and the autoclave curing process, which is necessary due to the high pressure and temperature requirements. This high manufacturing cost could be overcome through the application of ultraviolet (UV) curing technology. In the UV curing process, the resin only cures when UV radiation is applied [6]. This means that the UV prepreg storage does not require special conditions, such as in the case of traditional prepregs.

The UV curing leads to composite manufacturing in a faster timescale. However, this curing technique is usually limited to thin composite structures, because thicker parts require a longer curing period to fully penetrate the radiation, thereby reducing the efficiency of the production [7]. Moreover, it is reported that UV-curable composites suffer from low conversion due to the low penetration depth of the radiation as a result of the absorption of light by fibres and fillers, resulting in poor mechanical properties [8,9]. Thus, it is a controlled technology when in the field of coatings or thin films, but it needs adjustments to be applied to the curing of thick materials [10]

For these reasons, a new manufacturing process based on UV-cured pre-impregnated materials (UV prepreg) has been developed. The process involves two different stages. In the first stage, the UV prepregs are manufactured with a suitable resin content and with a sufficient degree of curing to avoid deimpregnation, giving rise to a prepreg with a high degree of tack and rollability. The laying up of the prepregs and the UV curing under pressure takes place in the second stage.

UV curing is widely used in high-productivity sectors, where a thin polymer layer is applied to a small and flat substrate, such as polymeric coatings, printing inks, and adhesives [11,12]. However, the high thicknesses and complex geometries used have hindered the application of the UV curing process for composites. The development of new photo-initiators for high wavelengths (360-405 nm) and new UV lamps could allow the curing of thicker polymer layers [13]. It could also give rise to the development of UV curing based on composite manufacturing technology. The maximum cured thickness [14], curing time [15], the optimum number of photo-initiators [15], the influence of post-curing (heat and time) [16], and influence of fibre in the curing behaviours of resins [15] have also been analysed in the recent years. Other new methods have been studied, such as cost-effective in situ UV curing [17], pultrusion curing [15,16,18], and automated fibre placement (AFP) [19–21] using UV technology.

One of the most important aspects to assess the final properties of composites is to control the degree of curing. In recent decades, several methods have been developed to study, control, and monitor the degree of curing [22–26]. Among these, dielectric analysis (DEA) [27,28] is one of the most suitable methods. It has gained significance in recent years in terms of monitoring the degree of curing for resins and composites, and it is employed in particular, to monitor the degree of curing in real-time situations [29–32]. In recent years, DEA technology has continued to develop with the

addition of new sensors [33,34]. It can be applied to different systems, such as thermoplastic infusion systems [35], epoxy prepreg systems [36], and even photopolymerized epoxy/acrylic systems [26].

This paper presents a new manufacturing process for a glass-fibre-reinforced composite based on UV-cured prepregs. In this study, the curing degree of the prepregs, related to the exposure time, is calculated from measured dielectric parameters. The effect of the fibre content of prepregs, the UV exposure time, the compaction pressure contribution on the fibre content of the composites, and the interlaminar shear strength are analysed to understand the flexural behaviour of the composites. The properties of the composites manufactured using this new process are compared to the properties of composites produced using traditional processes like hand lay-up and infusion.

## 2. Experimental

### 2.1. Materials

The composite manufactured in this study was a glass-fibre-reinforced acrylic polyester. The reinforcement was composed of three types of chopped strand mat (225, 300, and 450 g/m$^2$) and a woven fabric combined with a chopped strand mat (500 g/m$^2$), supplied by 3B-the fibreglass Company, noted as MAT225, MAT300, MAT450, and COMBI500, respectively. These reinforcements were used in the prepreg configuration. The reinforcement of any manufactured composite was composed of one layer of each material. The UV resin was a blend of two UV-curable polyester resins (TES21100 and VTC50) and an acrylic monomer (VTC5), supplied by IVM Chemicals. The photo-initiator was bis(2,4,6-trimethylbenzoyl)-phenylphosphineoxide, with the commercial name Irgacure® 819, provided by BASF the Chemical Company.

The conventional polyester resin used in the hand lay-up and infusion processes was Polylite® 413-575, supplied by Reichhold. Figure 1 displays the chemical reaction between the different species involved in the UV curing process.

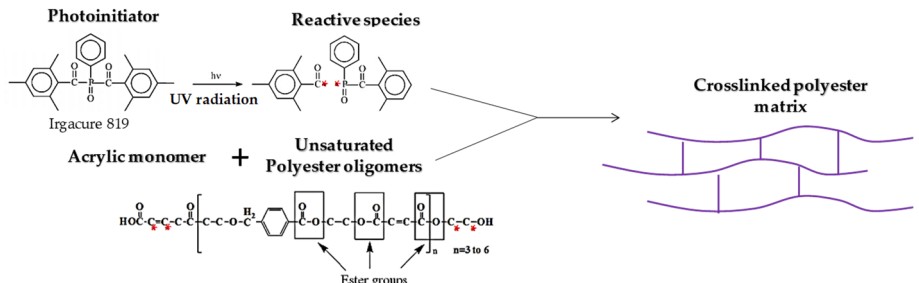

**Figure 1.** Schematic representation of the chemical reaction in the UV radiation curing process.

### 2.2. Prepreg Manufacturing Process

The prepreg manufacturing process consists of a fibre impregnation process, the adjustment of the resin content, and the precuring process. The impregnation process was carried out with hand rollers. To adjust the resin content of the prepregs to obtain the required resin content, we removed the excess resin with absorbent paper. The required value was measured by weighing the prepreg. The precuring process was carried out using a UV FUSION 330 conveyor belt, and a Hönle UVAHAND LED lamp with an emitting window of 137 × 75 mm at 120 mm from the sample, with an intensity of 50 ± 5 mW/cm$^2$ at a wavelength of 405 nm. The intensity of the LED lamp was measured using a Hönle UV meter. Both faces of the prepregs were exposed to obtain symmetrical precuring. Two different exposure times (0.25 s and 0.56 s for each face of each prepreg) were used to evaluate the influence of these parameters on the behaviour of the final composite.

The surface density of each reinforcement was different (225, 300, 450, and 500 g/m$^2$). Therefore, it was necessary to calculate the fibre content of each prepreg to give rise to a stack of prepregs, formed

by one layer of each reinforcement, resulting in the desired fibre content. Table 1 shows the fibre content of each prepreg to achieve a given fibre content for a stack of prepregs (20, 30, and 40 wt.%).

**Table 1.** Fibre content of the prepregs and the stacks of prepregs.

| Material | Fibre Weight (g) | Prepreg Weight (g) | Prepreg Fibre Content (%) |
|---|---|---|---|
| MAT225 | 2.43 | 16.11 | 15 |
| MAT300 | 3.45 | 19.35 | 18 |
| MAT450 | 5.04 | 24.78 | 20 |
| COMBI500 | 8.68 | 39.33 | 22 |
| Stack of Prepregs | 19.60 | 99.57 | **20** |
| MAT225 | 2.37 | 13.23 | 18 |
| MAT300 | 3.28 | 13.16 | 25 |
| MAT450 | 4.96 | 17.39 | 29 |
| COMBI500 | 7.95 | 18.87 | 42 |
| Stack of Prepregs | 18.56 | 62.65 | **30** |
| MAT225 | 2.69 | 12.45 | 22 |
| MAT300 | 3.13 | 7.98 | 39 |
| MAT450 | 4.52 | 9.56 | 47 |
| COMBI500 | 7.88 | 15.91 | 50 |
| Stack of Prepregs | 18.22 | 45.90 | **40** |

*2.3. Composite Manufacturing: Compaction and Curing Process*

The compaction process was carried out with an experimental set-up consisting of a metal surface and a metal frame. The prepregs and a glass sheet that was 100 mm wide, 100 mm long, and 10 mm thick were placed between both metal pieces. The metal frame was adjusted to the surface using four screws, pressing the prepregs uniformly through the glass. The homogeneous compaction was guaranteed by the stiffness of the 10 mm thick glass sheet being pressed throughout its perimeter. The torques of the screws were measured with a torque wrench. The compression stress was measured using a Kistler cell for different torques of the screws. The experimental set-up, the prepregs, and the final composite part can be seen in Figure 2, and the relationship between the pressure (bar) and torque (Nm) is shown in Figure 3. The displayed scale bars represent the standard deviation of three parallel experiments. Three different levels of pressure were applied with varying values of torque, namely, 0.75, 1.70, and 2.6 Nm, resulting in pressures of 2, 4, and 6 bar, respectively. Once the compaction pressure was achieved, the curing of the composites took place by switching on the same UV lamp and emitting window used for the prepreg manufacturing process, but now at the height of 35 mm from the glass plate.

To ensure the transmission of the UV light during the composite curing process, we performed an experimental procedure with no sample. This procedure consisted of measuring the intensity of the transmitted UV light in the presence and absence of the 10 mm thick glass plate. For this purpose, the Hönle UVHAND LED lamp and Hönle UV meter were used to generate and measure the transmitted radiation, respectively. In the experimental set-up, as in the composite manufacturing process, the UV light was placed 35 mm above the glass plate for 5 min. These results were compared with the results obtained for the same experiment but without the glass plate. Figure 4 shows the obtained results, where a small attenuation of 4% of the UV light intensity (from 183 to 175 ± 5 mW/cm$^2$) could be observed when the radiation passed through the glass plate. However, this was considered to be negligible for the experiment.

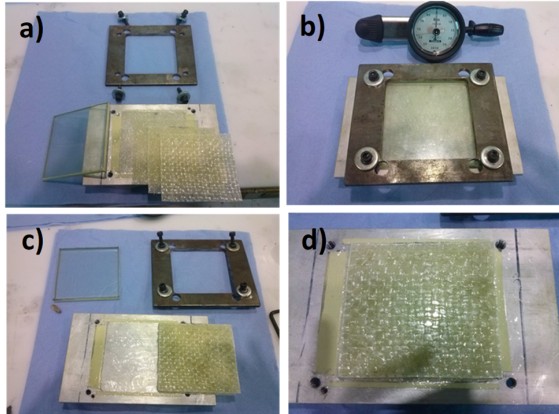

**Figure 2.** Experimental set-up for the compaction and curing of the final composite part. (**a**) prepregs layout; (**b**) composite manufacturing by compression step; (**c**) glass plate, metal frame, metal surface and composite layout; (**d**) final composite part.

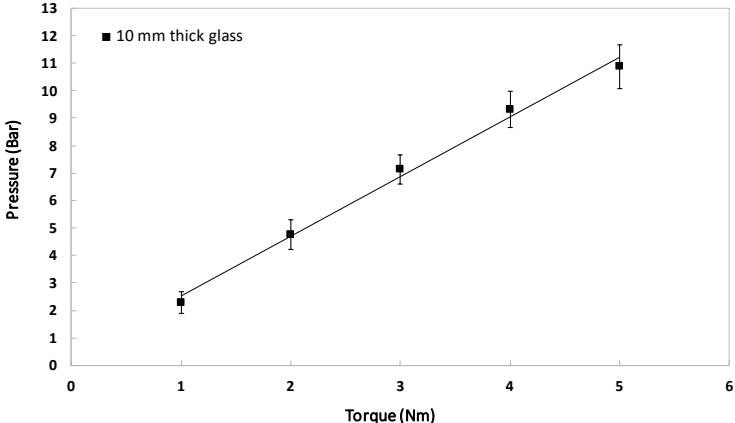

**Figure 3.** Pressure applied in the final composite vs. the torque used to grip the tool.

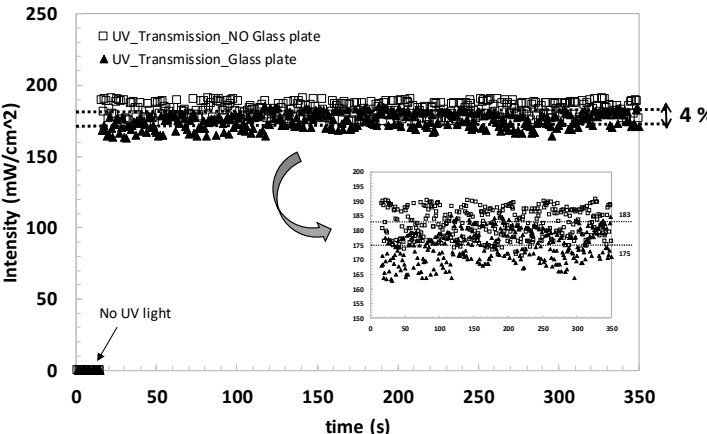

**Figure 4.** Recorded ultraviolet (UV) light intensity data vs. exposure time when the radiation passed through the 10 mm thick glass plate.

### 2.4. Composites Manufacturing Using the Hand Lay-Up and Infusion Processes

Composites with the same reinforcement as that used in the process based on the UV prepregs were manufactured using the hand lay-up and infusion processes to compare the new manufacturing process to the traditional manufacturing processes.

Regarding the hand lay-up process, the four layers of reinforcement, namely, MAT225, MAT300, MAT450, and COMBI500, were placed in a flat mould, and the resin was applied until the reinforcement was adequately wetted. The resin was cured at room temperature for eight hours.

Concerning the infusion process, the four layers of reinforcement were placed in a flat mould. The release layer and vacuum bag were placed at the top of the stack of the reinforcement. The vacuum bag was sealed at the perimeter with sealant tape, and a pipe was extracted to attach to the vacuum pump. Another pipe connected the sealed stack to the resin container. When the vacuum was applied, the resin wet the stack of the reinforcement and the air was removed. The resin cured at room temperature for eight hours.

### 2.5. Degree of Curing Measurement of the Prepregs and Composites

To analyse the degree of curing, we carried out dielectric analysis (DEA) using the NETZSCH DEA288 Epsilon curing monitor instrument.

DEA is based on tracking dielectric changes by applying an alternating electrical field to a sample. Interdigitated electrode sensors (IDEX) with a comb structure and an electrode distance of 115 μm are typically used for this process. Dipoles in the material attempt to orient themselves in the direction of the electric field and the charged species inside the sample are forced to move. This results in an electric field with a specific amplitude and phase change. The amplitude is related to the dielectric constant or dielectric permittivity, and the phase change is related to the dielectric loss factor. The permittivity reflects the number of dipoles in the material. The loss factor is the measurement of the total energy loss due to the work required to orient the dipoles and move the charged species. It is related to the ionic conductivity, which is the inverse of the ionic viscosity. The ionic viscosity is related to the degree of curing and is used to monitor the curing of thermoset polymers [27–37].

In this study, DEA analysis was used to measure the degree of curing of both the prepregs and composites. Concerning the curing of the prepregs, each uncured prepreg was placed on the top of the IDEX sensor, and the signal was recorded as a function of UV light exposure time. Concerning the composite parts, first, an experimental set-up was carried out to ensure the composite total curing. Figure 5 shows the schematic measurement representation, where each composite's interlayer was assessed. The experiment was divided into four separate measurements, wherein each experiment IDEX sensor was placed between the uncured prepreg layer, as shown in Figure 5. In all cases, DEA measurements began to be recorded before turning on the LED lamp.

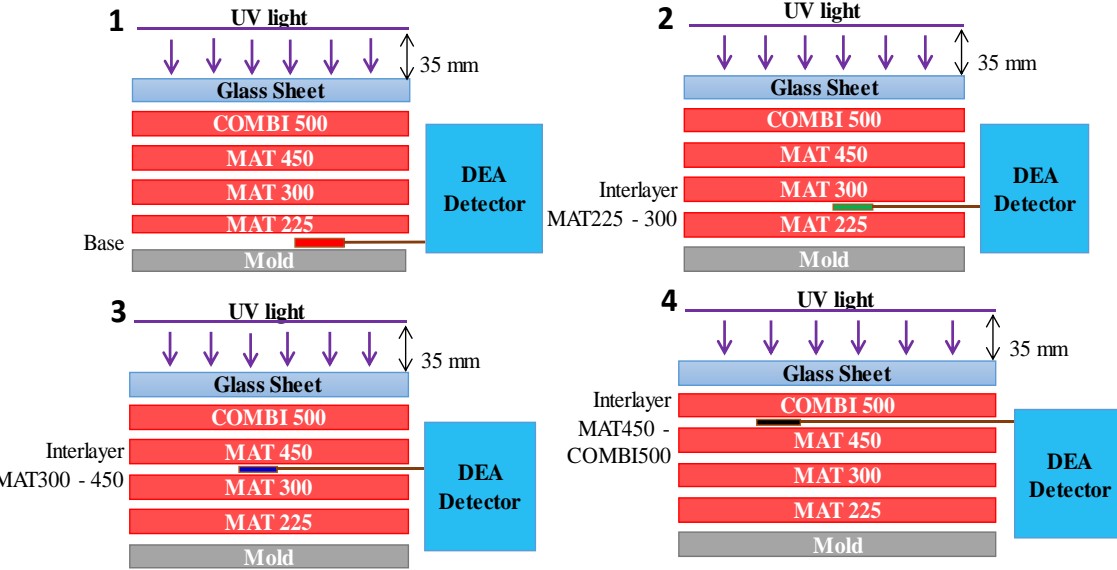

**Figure 5.** Schematic representation of the composite set-up interlayer curing degree measurement.

*2.6. Composite Fibre Content and Interlaminar Shear Strength (Ilss) Measurement and Flexural Behaviour*

The glass fibre weight fraction of the composites was measured by burning off the matrix in a muffle furnace. The matrix was removed by heating the samples to 600 °C for 6 h in preheated crucibles. The reinforcements were weighted after cooling down to room temperature in a desiccator, according to the ASTM D3171-15 standard.

The examinations of the interlaminar shear strength (ILSS) and flexural properties were carried out using the AG-X 5 kN universal testing machine by Shimadzu. The ILSS properties were tested according to the UNE EN ISO 14130:1999 standard, with a loading rate of 1 mm/min. The flexural properties were measured according to the ASTM D790-10 standard. The distance between two supporting points was set at 40 mm, and the loading rate was 1 mm/min. The samples were conditioned before the testing for 200 h at 23 °C and 50% RH (relative humidity). Both experiments were performed at room temperature. The resultant values were averaged with five parallel tests of the same samples.

## 3. Results and Discussion

As described previously, this new process consists of two different stages, namely, the manufacturing process of the UV prepregs and the compaction and curing processes of these prepregs to obtain the final composite part.

The manufacturing process of the prepregs consists of the impregnation, resin content adjustment, and precuring processes. The prepreg resin content adjustment and precuring processes could strongly affect the final fibre/resin ratio and delamination strength, and consequently, the final properties of the composite. The compaction pressure could also affect the final composite properties.

For these reasons, first, it is necessary to study the curing degree behaviours of the prepregs to relate them to the UV exposure time. Subsequently, the effect of the prepreg fibre content, the exposure time, and the compaction pressure on the composite fibre content and interlaminar shear strength were studied.

*3.1. Curing Degree Behaviour of the UV Prepregs*

As has been explained in the experimental part (Table 1), to obtain a stack of prepregs with the same reinforcement and different fibre contents (20, 30, and 40 wt.%), the fibre contents of the prepregs in each stack of the same chopped strand mat must be different. Thus, the same exposure time used to precure the prepregs of the same chopped strand mat could result in different curing degrees, considering that the more resin used, the longer it takes to cure if the transmission of the sample remains the same [38].

The curing degree of all the prepregs was analysed by DEA to check that prepregs of the same chopped strand mat with different resin contents had a different curing degree at the same exposure time.

The results of the DEA measurement were applied to obtain the dielectric loss factor. The loss factor and temperature during the curing process of the prepreg composed of MAT225 with 15 wt.% fibre content can be seen in Figure 6. As can be observed in the figure, the loss factor increases up to a maximum and then drops until it becomes constant. The loss factor is directly related to the amount of dissipated energy in molecular mobility [39]. Initially, the molecular mobility is favoured by the increase in temperature, so the loss factor increases initially. When the curing process progresses, the molecular mobility is restricted, causing a decrease in the loss factor. When the resin is fully cured, the loss factor remains constant [23,33,39]. From the loss factor, we could calculate the electrical conductivity. The inverse of the electrical conductivity is the ionic viscosity [39]. The temperature was measured by placing a thermocouple on the bottom of the sample.

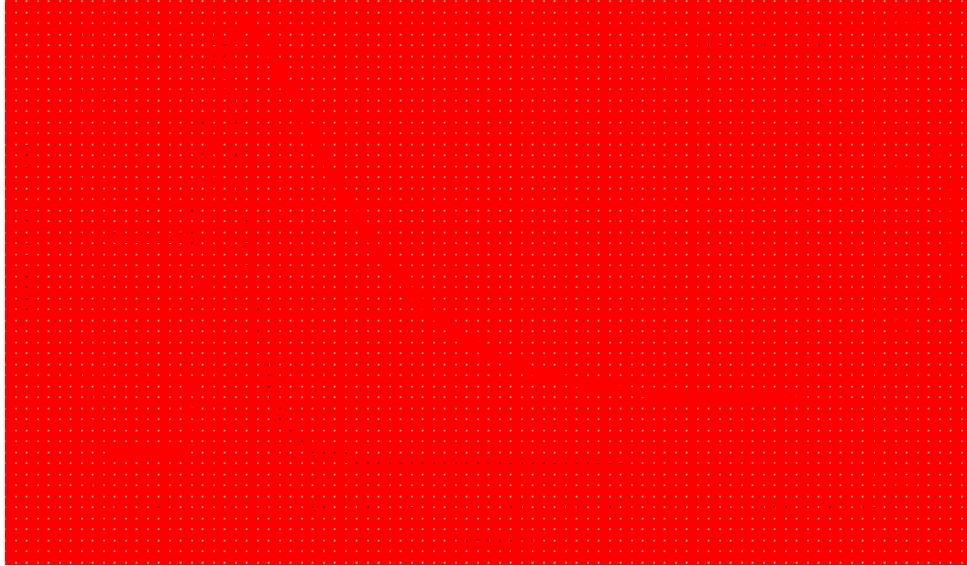

**Figure 6.** Loss factor (■) and temperature (●) in the curing process, analysed by dielectric analysis (DEA) of the UV prepreg composed of 15 wt.% MAT225.

Figure 7 shows both the ionic viscosity and temperature during the curing process of the prepreg composed of MAT225, with a 15 wt.% fibre content. When the reaction starts, a decrease in ionic viscosity takes place due to the increase in the temperature caused by the exothermic nature of the reaction. Then, as the reaction progresses, the ionic viscosity increases, because the energy necessary for the dipoles to move also increases. The ionic viscosity achieves a constant value when the reaction finishes [33,36]. Therefore, we can consider 0% conversion of the reaction when the ionic viscosity reaches the minimum value and 100% conversion when the ionic viscosity reaches the constant value. Thus, from the ionic viscosity curve, we could calculate the time-dependent curing degree [$\alpha(t)$] of the prepreg using Equation (1), where $\eta_t$ is the ionic viscosity at time $t$, $\eta_0$ is the initial viscosity of the resin, and $\eta_\infty$ is the maximum ionic viscosity of the resin [40,41].

$$\alpha(t) = (\log \eta_t - \log \eta_0)/(\log \eta_\infty - \log \eta_0) \tag{1}$$

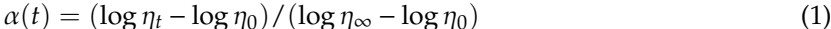

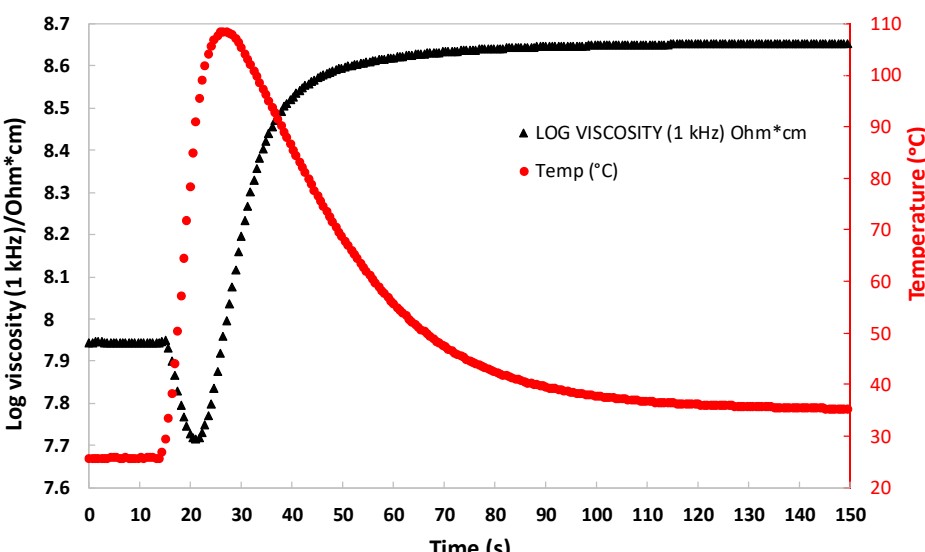

**Figure 7.** Log viscosity (▲) and temperature (●) in the curing process, analysed by DEA of the UV prepreg composed of 15 wt.% MAT225.

The curing degree behaviour of the prepreg of MAT225 with a 15 wt.% fibre content was calculated using Equation (1)—shown in Figure 8. The curing degree behaviour was calculated using the described method for all of the studied prepregs. The influence of the resin content in the degree of curing for the differently manufactured prepregs is discussed below.

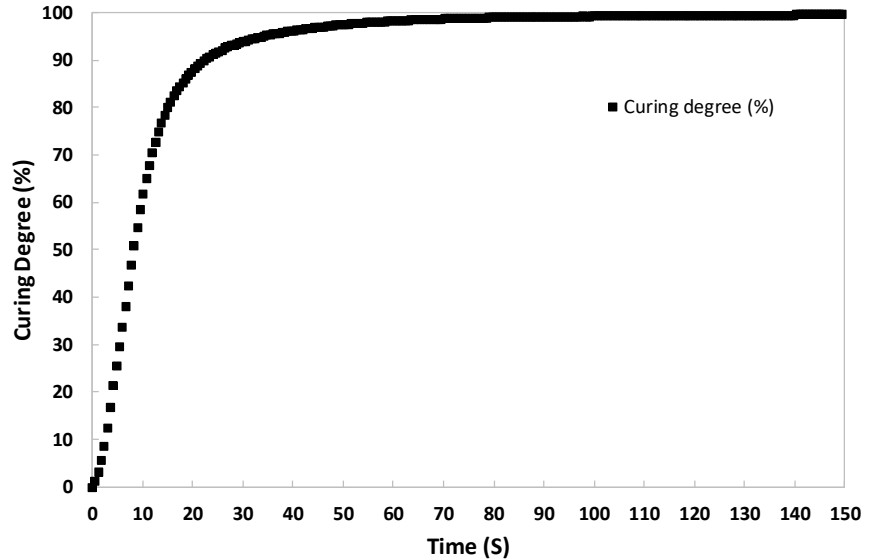

**Figure 8.** Curing degree (■) of the UV prepreg composed of 15 wt.% MAT225, calculated from parameters obtained by DEA.

Figure 8 shows the evolution of the curing degree for the MAT225 prepreg with fibre contents of 15 (■), 18 (♦), and 22 wt.% (▲). The left side of Figure 9 shows the slope of the curve with a 15 wt% fibre content that is lower than that with 18 and 22 wt.%. If we analyse the initial part of the curves, from 0 to 3 s of UV radiation (right side of Figure 8), at 1.5 s, the curing degree is less than 5% with a fibre content of 15 wt.% and around 10% with fibre contents of 18 and 22 wt.%. The displayed scale bars represent the standard deviation of three parallel experiments of each prepreg.

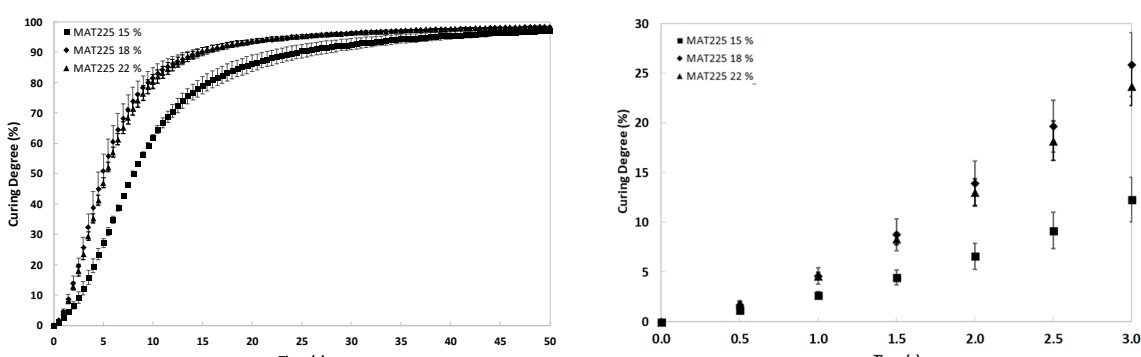

**Figure 9.** (**Left**) Curing degree of the MAT225 with a 15 (■), 18 (♦), and 22 wt.% (▲) fibre content. (**Right**) Magnification from 0 to 3 s time of curing.

This behaviour was similar in the case of the MAT300, where, at an exposure time of 1.5 s, the curing degrees were 4%, 7%, and 10% with 18, 25, and 39 wt.% fibre contents, respectively (Figure 10).

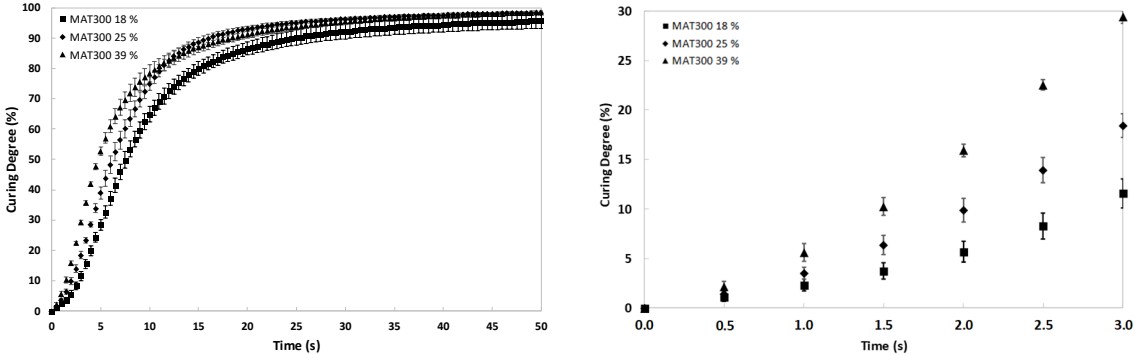

**Figure 10.** (**Left**) Curing degree of MAT300 with 18 (■), 25 (♦), and 39 wt.% (▲) fibre contents. (**Right**) Magnification from 0 to 3 s time of curing.

With respect to MAT450, the degree of curing was also higher when the fibre content increased. At an exposure time of 1.5 s, the curing degrees achieved were 2.5%, 6%, and 12% with 20, 29, and 47 wt.% (right side of Figure 11) fibre contents, respectively. Finally, with COMBI500, the difference in the degree of curing was lower than the standard deviation, despite the fact that the difference in the fibre content was higher, namely, from 22% to 42% and from 42 to 50 wt.% (right side of Figure 12).

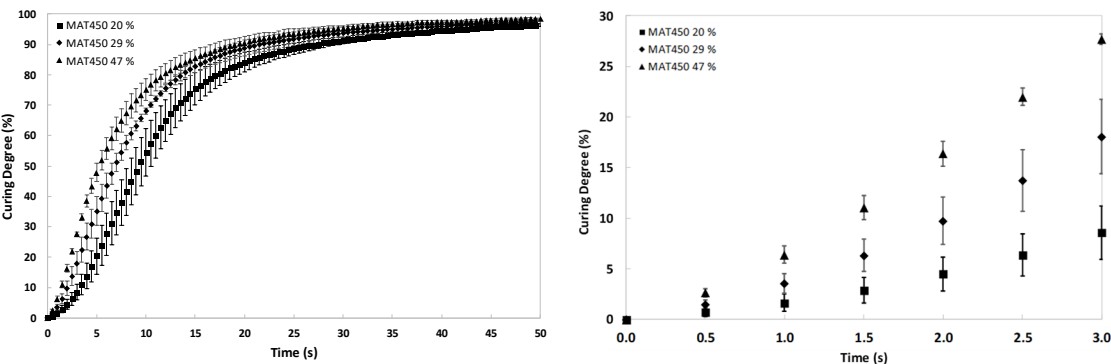

**Figure 11.** (**Left**) Curing degree of MAT450 with 20 (■), 29 (♦), and 47 wt.% (▲) fibre contents. (**Right**) Magnification from 0 to 3 s time of curing.

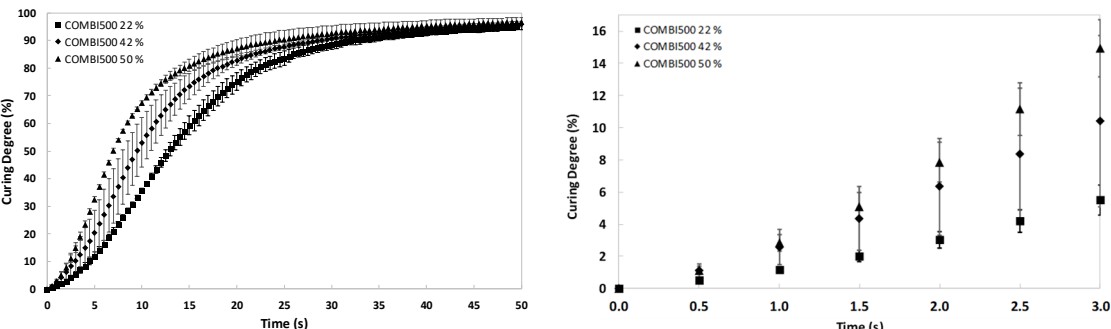

**Figure 12.** (**Left**) Curing degree of COMBI500 with 22 (■), 42 (♦), and 50 wt.% (▲) fibre contents. (**Right**) Magnification from 0 to 3 s time of curing.

These results demonstrated that the fibre content, in terms of the fibre/resin ratio, affected the relationship between the exposure time and the degree of curing. However, large differences in fibre content do not result in significant differences in the degree of curing. This effect is more pronounced in thicker fabrics, such as MAT450 and COMBI500, where DEA analysis was carried out on the surface of the prepregs because the sensor could not be used in the internal part of the prepregs. Consequently, the fibre content differences on the surface of each material were lower than that of the bulk material.

This indicated that it could be considered that the differences in the degree of curing of the UV prepregs of the same chopped strand mat with different fibre contents at the same exposure time were sufficient to affect the interlaminar adhesion and deimpregnation level.

### 3.2. Analysis of the Fibre Contents of the Composites

Figure 13 shows the fibre content of the composites manufactured from prepregs with fibre contents of 20 (■), 30 (♦), and 40 wt.% (▲), as explained in Table 1, with an exposure time of 0.56 s versus the compaction pressure. This figure presents that, for a pressure of 2 bar, the prepreg fibre content (20 wt.%) corresponds to a composite fibre content of 30 wt.%, whereas in the other two cases, the increase is less (from 30 to 32 wt.% and from 40 to 43 wt.%). The effect of pressure from 2 to 6 bar, in the case of samples manufactured with prepreg fibre contents of 30 and 40 wt.%, only results in increments of 6% (from 32 to 34 wt.%) and 12% (from 43 to 48 wt.%), respectively. This behaviour was completely different in the sample manufactured with a prepreg fibre content of 20 wt.%, which showed an increase in the final fibre content of the composite that was higher than 30% (from 30 to 40 wt.%) when pressure between 2 to 6 bar was applied. This was because of the effect of the exposure time of the prepregs on the UV light source. Prepreg fibre contents of 30 and 40 wt.%, i.e., resin contents of 70 and 60 wt.%, and an exposure time of 0.56 s generated a sufficient curing degree to avoid the excessive flow of the resin during the application of the compaction pressure. However, the achieved curing degree with an exposure time of 0.56 s and a prepreg fibre content of 20 wt.%, i.e., a resin content of 80 wt.%, was not enough to avoid deimpregnation at pressures higher than 4 bar.

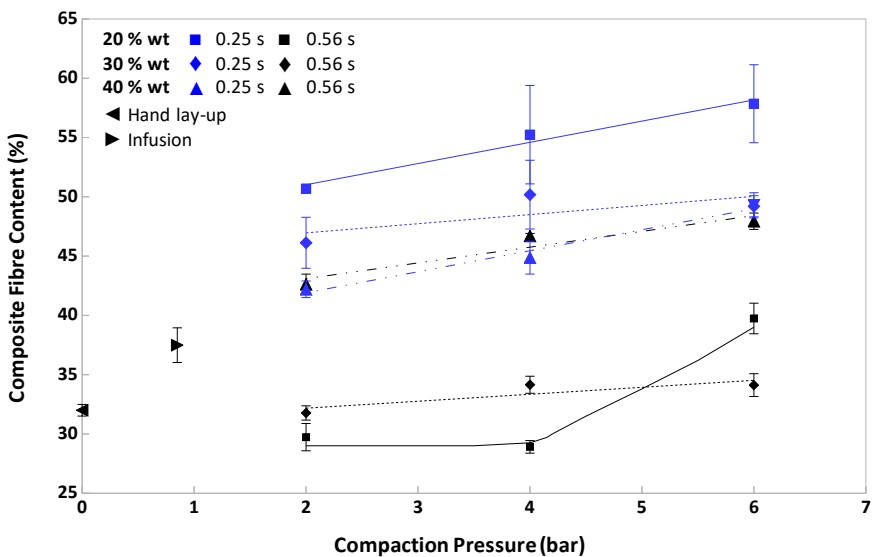

**Figure 13.** Fibre content of the composites with prepreg fibre contents of 20 (■), 30 (♦), and 40 wt.% (▲) and an exposure time of 0,56 s, and that of the composites with prepreg fibre contents of 20 (■), 30 (♦), and 40 wt.% (▲) and an exposure time of 0.25 s vs. the compaction pressure.

Figure 13 also displays the behaviour of the fibre content of the composites versus the compaction pressure with prepreg fibre contents of 20 (■), 30 (♦), and 40 wt.% (▲). However, at an initial exposure time of 0.25 s. In this case, only the sample with a 40 wt.% prepreg fibre content became a composite, with almost the same fibre content (from 40 to 42 wt.%) when a compaction pressure of 2 bar was applied. The prepreg fibre contents of 30 wt.% displayed an increase in the fibre content of the composite of 53% (from 30 to 46 wt.%) and the prepreg fibre content of 20 wt.% displayed an increase of 155% (from 20 to 51 wt.%). Increasing the compaction pressure from 2 to 6 bar resulted in an enlargement of the composite fibre content of 14% (from 51 to 58 wt.%), 7% (from 46 to 49 wt.%), and 16% (from 42 to 49 wt.%) for samples containing prepreg fibre contents of 20, 30, and 40 wt.%, respectively. These results indicated that for prepreg fibre contents of 20 and 30 wt.%, at an exposure time of 0.25 s, a slow

curing degree was achieved and deimpregnation processes appeared, especially when the pressure increased from 0 to 2 bar, producing composites with higher fibre contents. Nevertheless, an exposure time of 0.25 s, resulted in a curing degree that was sufficient to avoid de-impregnation when the prepreg fibre content was at approximately 40 wt.%.

Furthermore, Figure 13, as a reference, displays the composite fibre contents of 32 and 38 wt.%, manufactured by the hand lay-up (◄) and infusion (►) processes, respectively. These values were similar and higher than the value achieved with a prepreg fibre content of 20 wt.% with an exposure time of 0.56 s and compaction pressure equal to or lower than 4 bar. Moreover, when a higher compaction pressure was applied, i.e., 6 bar, a similar result was found to that obtained in the infusion process. For the rest of the results, that is, for 40 wt.% at an exposure time of 0.56 s and 20, 30, and 40 wt.% at an exposure time of 0.25 s, the achieved values overcame the results obtained for the hand lay-up and infusion processes.

### 3.3. Interlaminar Shear Strength (ILSS) Analysis

Figure 14 shows the interlaminar shear strength of the composites manufactured with prepreg contents of 20 (■), 30 (♦), and 40 wt.% (▲), at an initial exposure time of 0.56 s versus the compaction pressure. The same property of the composites manufactured by the hand lay-up (◄) and infusion (►) processes can also be observed in this figure. Values of the composites with a prepreg fibre content of 20 wt.% were higher than the rest of the values. This was due to the lower curing degree achieved in the prepregs, resulting in good adhesion between the laminates of the composites. The effect of the compaction pressure was only apparent in the composites with a 20 wt.% prepreg fibre content, and, in this case, the application of higher pressures, i.e., more than 4 bar, resulted in an increment of the interlaminar shear strength by 24% when compared to the value obtained for the infusion process (32.8 MPa vs. 26.4 MPa).

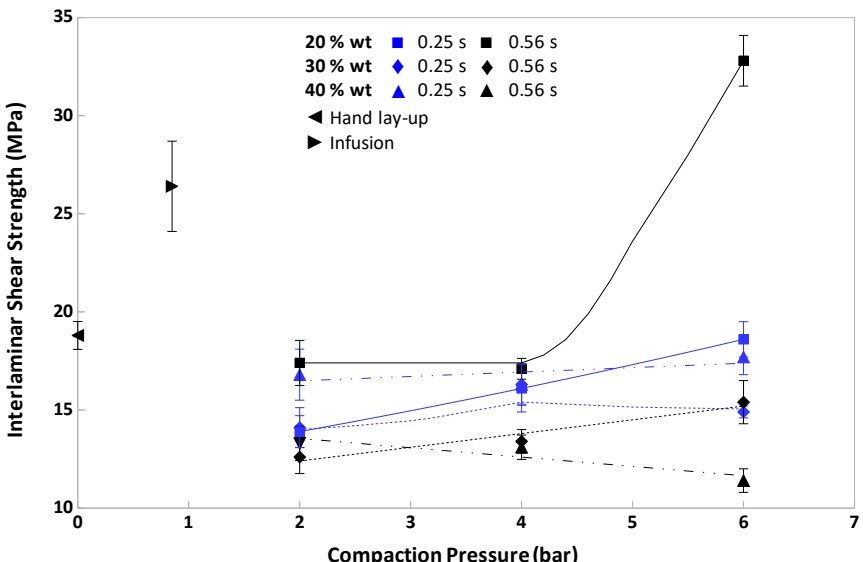

**Figure 14.** Interlaminar shear strength of the composites with prepreg fibre contents of 20 (■), 30 (♦) and 40 wt.% (▲) and an initial exposure time of 0.56 s. Additionally, those with prepreg fibre contents of 20 (■), 30 (♦), and 40 wt.% (▲) and an initial exposure time of 0.25 s vs. the compaction pressure.

The increased interlaminar shear strength could be due to the compaction forces that increase the infiltration of the liquid matrices between adjacent layers [15]. To verify this behaviour, the stress–displacement curves for the composites with prepreg fibre contents of 20 (■) and 30 (♦) wt.% and an initial exposure time of 0.56 s, both at a pressure of 6 bar, are shown in Figure 15. Only a representative curve of one of the five test samples is displayed. As can be seen in Figure 15,

delamination in the composite with a 30 wt.% prepreg fibre content at an exposure time of 0.56 s appeared at approximately 15 MPa. However, this shear stress was not enough to delaminate the composite sample with a 20 wt.% prepreg fibre content at 0.56 s, where delamination appeared at shear stress values close to 30 MPa. In this case, the compaction forces (6 bar) increased the infiltration of the liquid matrix between adjacent layers, and because the degree of curing of the matrix was less at the same exposure time, the amount of resin was greater in the case of the 20 wt.% prepreg fibre content (80 wt.% resin content) versus the 30 wt.% prepreg fibre content (70 wt.% resin content).

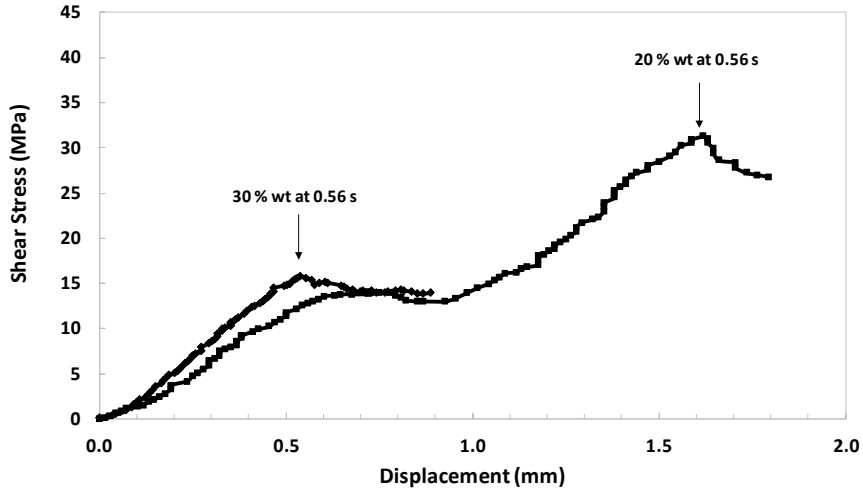

**Figure 15.** Shear stress–displacement curves for composites with prepreg fibre contents of 20 wt.% (■) and 30 wt.% (♦) with an initial exposure time of 0.56 s.

The interlaminar shear strength behaviour of the composites manufactured with prepreg fibre contents of 20 (■), 30 (♦), and 40 wt.% (▲), at an initial exposure time of 0.25 s, are also displayed in Figure 14. All the obtained values were higher than values with 30 and 40 wt.% prepreg fibre contents at an exposure time of 0.56 s. This result indicated that the low curing degree of the prepregs decreased the delamination effects.

As a way to improve other industrial processes, there have been studies in recent years on the interlaminar shear strength of other composites based on glass fibre and UV-curing resins [15,19–21], including pultrusion [16] and AFP [19–21]. The ILSS achieved values in processes developed to improve AFP of around 12 MPa [19,20] and between 20 and 30 MPa when a force of 200 N was applied [21]. These values agree with the ones obtained in this study, even though the reported values were obtained using fabrics in a woven configuration. In the present work, approximately 70% of the fabrics were in a MAT configuration. In processes related to pultrusion, these values are much higher, i.e., from 34 to 44 MPa, but all the layers are impregnated at the same time [15].

Concerning other composites based on glass fibre, the interlaminar shear strength values are similar using as a resin a thermal curing polyester (from 5.6 MPa to 32.4 MPa) [42], slightly higher using epoxy resin (33.06 MPa) [43], and much higher using modified epoxy resin (41.46 MPa) [43]. In all cases, the glass fabric is in the woven configuration.

### 3.4. Flexural Behaviour

Figure 16 shows the flexural moduli of the composites with prepreg fibre contents of 20 (■), 30 (♦), and 40 wt.% (▲) at an initial exposure time of 0.56 s versus the compaction pressure. The flexural moduli of the composites manufactured using the hand lay-up (◄) and infusion (►) processes can also be seen in this figure. The highest flexural modulus values corresponded to composites manufactured with a 40 wt.% prepreg fibre content. The values of the modulus for the composites manufactured with a 20 wt.% prepreg fibre content were higher than values with a 30 wt.% content. In all cases,

the modulus values increased with the compaction pressure. However, the increment was higher for composites manufactured with a 40 wt.% prepreg fibre content (21%) than that of the composites manufactured with prepreg fibre contents of 20 wt.% (11%) and 30 wt.% (6%), respectively. However, the tendencies were not clear, considering the high errors bars of the values. These slightly higher values of the modulus in the composites manufactured with a prepreg fibre content of 40 wt.% were due to the composite fibre content, which was much higher than 40%, as shown in Figure 13, which also increased with pressure.

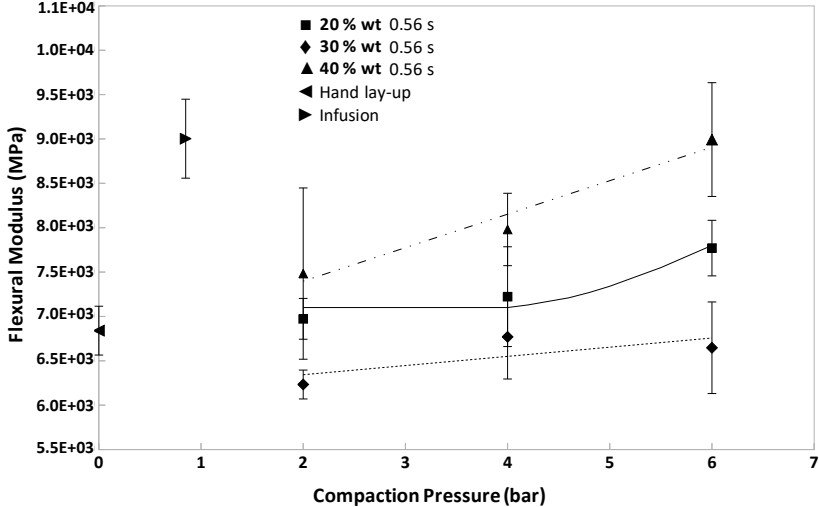

**Figure 16.** Flexural moduli of the composites with prepreg fibre contents of 20 (■), 30 (♦), and 40 wt.% (▲) and an initial exposure time of 0.56 s vs. the compaction pressure.

The flexural modulus behaviour of the composites manufactured with 20 and 30 wt.% prepreg fibre contents agree with the composite fibre content behaviour observed in Figure 12. At low and medium pressures (i.e., 2 and 4 bar), the modulus values were similar, and at higher pressures (i.e., 6 bar), the composites manufactured with a 20 wt.% prepreg fibre content had a higher modulus than the composites manufactured with a 30 wt.% prepreg fibre content.

Comparing the modulus values of the new UV curing process studied in this work with the values of traditional processes, such as the infusion and hand lay-up processes, the modulus behaviour of the latter process was comparable to the composite fibre content behaviour obtained by UV curing. The fibre content of the composites manufactured using the hand lay-up process (32 wt.%) was similar to the values of the composites manufactured using the UV process with prepreg fibre contents of 20% and 30 wt.%, and lower than the values of the composites with a prepreg fibre content of 40 wt.%. Concerning the modulus values of the composites manufactured using the infusion process, they were similar to the highest modulus values of composites manufactured using this new process, even though the composite fibre content was lower. This result was most likely due to the infusion resin behaviour, which was a commercial resin optimized for this process. In contrast, the resin used in this study was an experimental acrylic UV resin.

The same parameters as in Figure 16, but with an initial exposure time of 0.25 s, can be observed in Figure 17. All the values at this exposure time are higher than those obtained for 0.56 s. The result was because of the higher composite fibre content obtained for composites manufactured at an exposure time of 0.25 s, where the curing degree of the resin is lower, and deimpregnation effects appear, as discussed previously. The impact of compaction pressure does not influence low pressures. However, the modulus values increase when the pressure rises from 4 to 6 bar. This increment is sharper than the fibre content increase seen in Figure 13. It is reported that in other glass fibre reinforced composites; there is an increment of the Young modulus when the interfacial adhesion is improved [44,45].

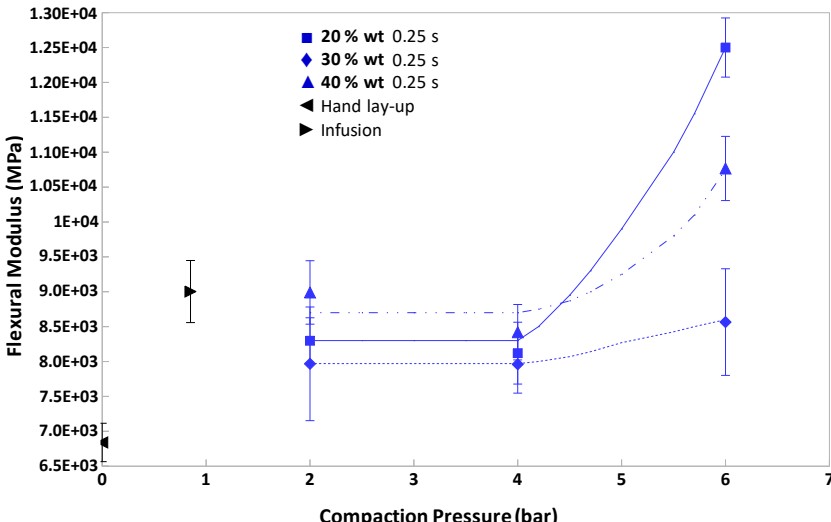

**Figure 17.** Flexural moduli of the composites with prepreg fibre contents of 20 (■), 30 (♦), and 40 wt.% (▲) and an initial exposure time of 0.25 s vs. the compaction pressure.

Given these results, there was a need to apply high pressures to the composite to reduce the possible voids between the different laminates, giving rise to better surface contact between the resin and fibres.

Figure 18 shows the flexural strength of the composites with prepreg fibre contents of 20 (■), 30 (♦), and 40 wt.% (▲) at an initial exposure time of 0.56 s versus the compaction pressure. The flexural strength of the composites manufactured using the hand lay-up (◄) and infusion (►) processes can be seen in this figure. The values of the flexural strength of the composites manufactured with a prepreg fibre content of 20 wt.% were much higher than the composites with 30 and 40 wt.%. This was due to the exposure time, and consequently, the curing degree in the prepregs, which resulted in delamination effects. The curing degree was too high for the prepreg fibre contents of 30 and 40 wt.%. In all cases, the compaction pressure must be higher than 4 bar to increase the flexural strength, indicating that a compaction pressure higher than 4 bar promotes interlayer adhesion effects [15].

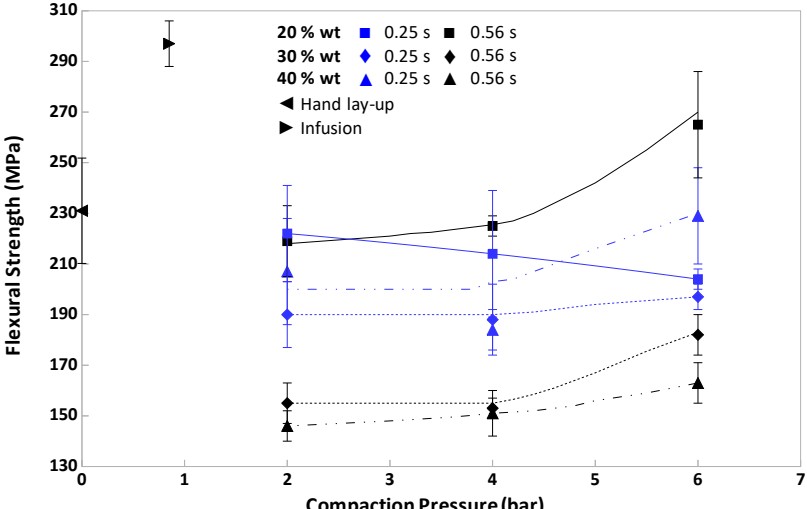

**Figure 18.** Flexural strength of the composites with prepreg fibre contents of 20 (■), 30 (♦), and 40 wt.% (▲) and an initial exposure time of 0.56 s, and those with prepreg fibre contents of 20 (■), 30 (♦), and 40 wt.% (▲) and an initial exposure time of 0.25 s vs. the compaction pressure.

Moreover, Figure 18 displays values of the flexural strength of the composites manufactured with prepreg fibre contents of 20 (■), 30 (♦), and 40 wt.% (▲) at an exposure time of 0.25 s. All values were much higher than that achieved for the composites manufactured with an exposure time of 0.56 s and prepreg fibre contents of 30 and 40 wt.%. This indicated that delamination effects were less important at lower exposure times in samples with these prepreg fibre contents. The effect of the compaction pressure for samples with 30 and 40 wt.% prepreg fibre contents at an exposure time of 0.25 s was similar to that obtained for an exposure time of 0.56 s. That is, a compaction pressure higher than 4 bar is necessary to increase the flexural strength. However, for prepregs with a 20 wt.% fibre content and an exposure time of 0.25 s, a low curing degree was achieved, and deimpregnation effects appeared at higher compaction pressures, provoking a decrease in the flexural strength.

The flexural strength behaviour of the composite does not correspond precisely to the ILSS behaviour. This could be due to the insufficient curing when considering the entire volume of the composite. Figure 19 shows the ionic viscosity of each layer of a composite, compressed at a pressure of 2 bar. The prepreg fibre content of this composite was 20 wt.%. The measurement was made in this composite because it was the thickest, so it was the worst situation for the penetration of UV light. The total curing of the rest of the composites (30 and 40 wt.%) could be supposed if the measured composite was cured completely.

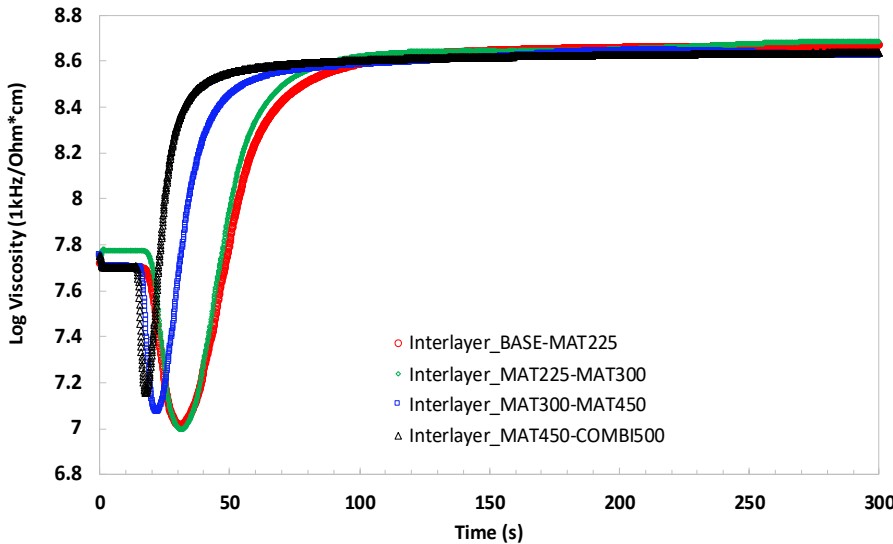

**Figure 19.** Log viscosity of a UV-cured composite with a 20 wt.% fibre content pressed at 2 bar.

Each curve corresponds to the curing behaviour of the bottom surface of the different layers composed of MAT225 (15%), MAT300 (18%), MAT450 (20%), and COMBI500 (22%). The maximum value of the ionic viscosity measured in all the layers was the same here. This result indicated that all the layers achieved the same curing degree and were considered to reach 100% because from 150 s to 300 s, the ionic viscosity remained constant in all the layers.

We compared the values of the flexural strength of the new process studied in this work with the values of traditional processes, such as the infusion and hand lay-up processes. We observed that the behaviour of the flexural strength was in agreement with the interlaminar shear strength (see Figure 14) of the hand lay-up process (19 MPa). The flexural strength value of the hand lay-up process was similar to the highest values obtained for the composites manufactured using the UV process between 2 and 4 bar. However, it was lower than the value of the composites with a prepreg fibre content of 40 wt.% at a pressure of 6 bar. Regarding the flexural strength values of the composites manufactured by the infusion process, these were higher than the highest values of the composites manufactured by the UV processes, although the interlaminar strength was lower. This behaviour pointed towards the properties of the infused resin.

## 4. Conclusions

The results of this study indicated that substantial differences in the fibre contents of all the prepregs considered do not result in significant differences in the degree of curing, as calculated from dielectric parameters measured by DEA on the surface of the prepregs, especially in thicker materials.

The effect of the compaction pressure on the fibre content of composites at an exposure time of 0.56 s is weak at medium and high prepreg fibre contents, and it is strong at a low prepreg fibre content. When the exposure time decreased to 0.25 s, this effect was less at a high prepreg fibre content, whereas the effect was stronger for medium and low prepreg fibre contents, even at low pressures.

The interlaminar adhesion was low and not very sensitive when the compaction pressure was applied to medium and high prepreg fibre content samples, obtained at an exposure time of 0.56 s. However, this property increased considerably at the same exposure time when both, a low prepreg fibre content and a high pressure (i.e., more than 4 bar), were used. The interlaminar adhesion at these conditions was superior to the values obtained from the infusion process. At an exposure time of 0.25 s, the interlaminar adhesion increased significantly. However, it did not achieve the highest values reached at longer exposure times.

The behaviours of flexural moduli of composites reported in this study are in general agreement with that of the fibre content of the composites. The flexural strength values were low, except in the conditions in which interfacial adhesion was promoted. This effect was particularly evident in the composite that was prepared with a prepreg fibre content of 20 wt.% at a pressure higher than 4 bar. The low flexural strength values of the rest of the samples are not due to insufficient curing but rather low interfacial adhesion.

The results presented in this work indicate that this new UV process development is very promising, as the properties of the composites manufactured with favourable parameters were better than the properties of composites manufactured using traditional methods. However, further research regarding the optimization of the properties of the UV resin and ideal process is necessary.

**Author Contributions:** Conceptualization, F.J.V.; methodology, A.A.I. and N.G.P.-d.-E.; software, K.G.; validation, N.G.P.-d.-E., A.A.I. and F.J.V.; formal analysis, N.G.P.-d.-E., A.A.I. and F.J.V.; investigation, N.G.P.-d.-E., A.A.I. and F.J.V.; resources, K.G. and F.J.V.; data curation, N.G.P.-d.-E. and A.A.I.; writing—original draft preparation, A.A.I. and F.J.V.; writing—review and editing, N.G.P.-d.-E. and F.J.V.; visualization, K.G.; supervision, F.J.V.; project administration, F.J.V. and K.G.; funding acquisition, F.J.V. and K.G. All authors have read and agreed to the published version of the manuscript.

**Funding:** The authors disclose receipt of the following financial support for the research, authorship, and/or publication of this article: This work was supported by the Basque Agency for Business Development (SPRI) (project number ZL-2018/00260).

**Conflicts of Interest:** The authors declare that they have no conflict of interest.

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
