# Peer review of "Influence of the Fibre Content, Exposure Time, and Compaction Pressure on the Mechanical Properties of Ultraviolet-Cured Composites"

_jcs, doi:10.3390/jcs4010030_

Round 1
Reviewer 1 Report
This paper investigates the effects of UV curing of various prepregs fabricated by the authors. The authors have compared different compositions of prepreg exposed to varying time to the UV radiation for flexure strength and other relevant properties.
This paper is technically significant, however, it needs thorough proofreading before it is accepted for publication. I am highlighting a few of the concerns I have below:
1. Line 29: "High property requirement"- Please list the properties you are referring.
2. Line 31: rewrite this sentence.
3. Figures 8, 9, 10 and 11: can you separate the subplot in these figures.
Author Response
The manuscript's english languge has been edited in the MDPI English service.
For the responses, Please see the attachment

Reviewer 2 Report
Reviewers comments:
The article based on title “Influence of the Fibre Content, Exposure Time and Compaction Pressure on the Mechanical Properties of Ultraviolet Cured Composites” reported by Pérez-de-Eulate et al. is well written and presented. But, still so many quarries need solution for publication.
- Abstract should be quantitative. Some technical part should insert. Introduce some more papers for improving introduction. Carbohydrate polymers 211, 181-194. Nanomaterials 9 (11), 1523
- Introduction should be updated by 2019. Check and modify it. Novelty of the research should be mention in introduction.
- All the provided figure needs scale bar code. Check and modify it. Fig. 8,9,10,11 contained inset should clear with proper explanation.
- Reported similar work needs comparison with present work. Flexural and shear stress behaviour
- Line 402-404 needs proper references.
- The flexural modulus present at Y-axis in Fig 16 needs in power of tenth. Check it and modify.
- What about scheme for such reported materials. Improve the given and clear it.
- Line 191….needs modified like….The values of results were the averaged with five parallel tests of the same samples.
Author Response
The manuscript's english language has been edited in the MDPI English service.
Please see the attachment.

Round 2
Reviewer 2 Report
Accept
This manuscript is a resubmission of an earlier submission. The following is a list of the peer review reports and author responses from that submission.
Round 1
Reviewer 1 Report
This the second review of the manuscript. Although the authors claim that the intensity of the UV LED lamp was measured through the glass used in the manufacturing process, it is not clear what they mean and how this measurement was performed. Next to that the authors did not comment at all to the following point "Thus the parametric study conducted here is more appropriate for a technical report and does not bring new scientific insight". These are major issues that were not appropriately addressed in the revised paper.
Reviewer 2 Report
I suggested rejecting this manuscript in the first review. The revised manuscript was only slightly improved. However, the manuscript still has poor results and cannot be improved.Reviewer 3 Report
It doesnt make sense that UV ray penetrate through the thick composite to the bottom lay. Also the viscosity is not measurable if the epoxy is fully cured I think. My one and simple query is not resolved well in this revision. Thus this reviewer suggest reject.